# Therapeutic efficacy of Chloroquine for the treatment of uncomplicated *Plasmodium vivax* infection in Shewa Robit, Northeast Ethiopia

Habtamu Belay[1]*, Megbaru Alemu[2☯], Tadesse Hailu[2], Hussein Mohammed[3☯], Heven Sime[3], Henok Hailegeorgies[3], Bokretsion Gidey[3], Mebrahtom Haile[4], Gudissa Assefa[4], Worku Bekele[5], Mihreteab Alebachew Reta[6], Andargachew Almaw Tamene[7], Geremew Tasew[3☯], Ashenafi Assefa[3,8]

1 Department of Medical Laboratory Science College of Medicine and Health Science, Wolkite University, Wolkite, Ethiopia, 2 Department of Medical Laboratory Science, College of Medicine and Health Sciences, Bahir Dar University, Bahir Dar, Ethiopia, 3 Ethiopian Public Health Institute, Addis Ababa, Ethiopia, 4 Ethiopian Ministry of Health, Addis Ababa, Ethiopia, 5 World Health Organization Addis Ababa, Ethiopia, 6 Department of Medical Laboratory Science, College of Medicine and Health Sciences, Wollo University, Dessie, Ethiopia, 7 Department of Medical Laboratory Science, College of Health Sciences, Debre Tabor University, Debre Tabor, Ethiopia, 8 Institute for Global Health and Infectious Disease, University of North Carolina at Chapel Hill, Chapel, North Carolina, United States of America

☯ These authors contributed equally to this work.
* belayhabtish2424@gmail.com

**Data Availability Statement:** All relevant data are within the paper and its Supporting information files.

## Abstract

### Background

The development of drug resistance to chloroquine is posing a challenge in the prevention and control efforts of malaria globally. Chloroquine is the first-line treatment for uncomplicated *P.vivax* in Ethiopia. Regular monitoring of anti-malarial drugs is recommended to help early detection of drug-resistant strains of malaria parasites before widely distributed. The emergence of *P.vivax* resistance to chloroquine in the country endangers the efficacy of *P.vivax* treatment. This study aimed to assess the therapeutic efficacy of chloroquine among uncomplicated *P.vivax* infections at Shewa Robit Health Center, northeast Ethiopia.

### Methods

One-arm *in vivo* prospective chloroquine efficacy study was conducted from November 2020 to March 2021. Ninety participants aged between 16 months to 60 years confirmed with *P.vivax* mono-infection microscopically were selected and treated with a 25 mg/kg standard dose of chloroquine over three days. Thick and thin blood smears were prepared and examined. Clinical examination was performed over 28 follow-up days. Hemoglobin concentration level was measured on days 0, 14, and 28.

### Result

Of the 90 enrolled participants, 86 (96%) completed their 28 days follow-up period. The overall cure rate of the drug was 98.8% (95% CI: 95.3–100%). All asexual stages and gametocytes were cleared within 48 hours with rapid clearance of fever. Hemoglobin

**Funding:** The authors received no specific funding for this work.

**Competing interests:** The authors have declared that no competing interest exists.

**Abbreviations:** ACPR, Adequate Clinical and Parasitological Response; AE, Adverse Events; CQ, Chloroquine; CQR, Chloroquine Resistance; DCQ, Desethylchloroquine; ETF, Early Treatment Failure; FMoH, Federal Ministry of Health; Hgb, Hemoglobin; LCF, Late Clinical Failure; LPF, Late Parasitological Failure; PRR, Parasite Reduction Ratio; WHO, World Health Organization.

concentration had significantly recovered between days 0 and 14, 0 and 28, and 14 and 28 days ($P = 0.032$, $P<0.001$, and $P = 0.005$), respectively. Fast resolution of clinical signs and symptoms was also observed. Severe adverse events were not recorded.

## Conclusion

The present study revealed that chloroquine remains an efficacious and safe drug in the study setting for treating uncomplicated *P.vivax* in the study area. Large-scale continuous surveillance is needed to monitor the development of resistance in due time.

## Background

Malaria is still a major cause of death and severe illness in many areas of the world. There were an estimated 241 million reported cases and 627 000; malaria-related deaths in 2020. Africa takes the biggest portion of cases and deaths of any continent ($\sim$ 96%) [1]. *Plasmodium vivax (P.vivax)* is the most prevalent human malaria parasite found in many parts of the tropical and subtropical regions of the world. About 4.5 million cases of *P.vivax* malaria were reported worldwide in 2020 [1]. The highest burden of *P. vivax* infection was reported throughout the countries in Southeast Asia and South America [2]. In sub-Saharan countries, the number of *P. vivax* cases was found sparsely distributed and becoming gradually increased [3].

In Ethiopia, two-thirds of the population (~53.5million people) lives in areas at risk of malaria infection and 26% of malaria cases were caused by *P.vivax* [4]. Gradual and heterogeneous increase in *P. vivax* cases has been observed in the past several years [5].

Chloroquine (CQ) is the first-line treatment for *P. vivax* mono-infection in Ethiopia [6]. However, after long-term extensive use, widespread resistance to CQ has been reported in many parts of the world including Ethiopia [7,8]. The first treatment failure due to chloroquine resistance (CQR) was documented in 1989 in Papua New Guinea [9].

In Ethiopia, the first report of CQ resistance was reported in Debre Zeit (4.6%) and consequently there were reports of failure in different parts of the country [10,11]; on the other hand, the majority of study results revealed that CQ is still sustained as an efficacious drug for *P.vivax* infection in the country [12,13]. Thus, a continuous monitoring and surveillance system is needed to monitor the therapeutic efficacy of CQ and to trace the development of CQR in due time. This study aimed to deliver updated information about the status of therapeutic efficacy of CQ for the treatment of uncomplicated *P.vivax* in Shewa Robit Health Center, northeast Ethiopia.

## Methods

### Study period, site, and population

This study was conducted between November 2020 to March 2021 at Shawa Robit Health Centre, located in Shawa Robit town administration, North Shawa Zone, Amhara Regional State, northeast Ethiopia. Shawa Robit is located at 1280 meters above sea level with; a longitude and a latitude of 10°00′N 39°54′E, 225 km northeast of Addis Ababa. According to the Shewa Robit health office data, the total catchment population of Shewa Robit health center in 2021 was estimated at around 60,234. The area receives high rainfall during the main rainy seasons (June to September) and the annual rainfall is about 1000 mm. The area is characterized by markedly unstable seasonal malaria. Malaria is one of the top ten diseases in the town and is

reported throughout the year [14]. The study participants were recruited from all malaria-suspected individuals attending the outpatient department of the study health center. Patients confirmed with uncomplicated *P.vivax* mono-infection who fulfilled the WHO inclusion criteria were enrolled in the study [15].

### Inclusion criteria

Patients aged ≥ 6 months, mono-infected with *P.vivax* confirmed by microscopic blood smear with asexual parasitemia > 250/μl of blood, body weight > 5 kg, non-pregnant or breastfeeding women, patients living within the health center catchment area (10km radius of the health center), axillary temperature ≥ 37.5 ˚C or who have a history of fever within the previous 48 hours, informed consent by the patient or by caregivers for children under 12 years old, agreed to return for all scheduled visits, and willing to comply with the study protocol were enrolled to the study [15].

### Exclusion criteria

Patients infected with other than *P.vivax* malaria species, presence of febrile condition other than malaria or known underling chronic or severe disease, presence of severe malnutrition defined by WHO, known hypersensitivity to the study drug, hemoglobin level who have < 5.0mg/dl, unable to take oral CQ medication or having continuous vomiting, patients who took antimalarial drugs within 2 weeks before enrolment and who had regular medication that may interfere with the study drug pharmacokinetics were excluded from the study [15].

### Study design and sample size

World Health Organization (WHO) 2009 guideline, methods for the surveillance of antimalarial drug efficacy was used as a reference standard [15]. A one-arm *in vivo* prospective study was designed. Sample size was determined by using the single population proportion formula and assuming a 5% margin of error, 95% confidence interval (CI), treatment failure of 5%, and an additional 20% loss to follow-up rate and withdrawal of consent [15]. Accordingly, 90 study participants were enrolled in the study.

### Baseline evaluation and data collection

Base-line physical and clinical examinations with particular attention to any danger signs or symptoms associated with severe malaria were thoroughly assessed by a clinician. Febrile patients were treated with an appropriate dose of paracetamol. Socio-demographic information from the study participants was collected and recorded on a standardized CQ efficacy case screening form [15] by well-trained clinicians and senior laboratory technologists using questionnaires. A Patient who meets the selection criteria at this stage was assigned a patient identification number and referred to the laboratory again for further laboratory investigation and sample collection.

### Treatment, dosing, and follow up

Enrolled patients were treated with the standard Chloroquine phosphate 250 mg coated tablet, (Manufacturer Rimedica Ltd Aharnon Str, Limassol Industrial Estate, 3056 Limassol, Cyprus, EU, batch number 80368 and expiry date 02/2024) weight-based (a 25 mg/kg) standard dose of chloroquine over three days; 10mg base/kg on Days 0 and 1, and 5mg base/kg on Day 2 was administered under direct observation of qualified clinicians. Drug dosage was determined according to the national malaria treatment guideline of Ethiopia [6].

The study patients were observed for 30 minutes after drug administration for vomiting. Any patient who vomited during the observation period was re-treated with the same dose and observed for an additional 30 min. If the patient vomits again, he or she was withdrawn and offered rescue therapy [15]. The follow-up consisted of a fixed schedule of check-up visits for a 28-day follow-up period using a standardized drug efficacy record form, on days 0, 1, 2, 3, 7, 14, 21, and 28. Primaquine 0.25 mg base per kg daily for 14 days was administered for radical cure after the study period as per the national malaria treatment guideline [6].

## Clinical evaluations

A standard physical examination, body weight, axillary temperature, and clinical conditions were examined and recorded on days 0 and days 1, 2, 3, 7, 14, 21and 28 days follow-up period.

## Laboratory procedures

**Microscopic blood film examination.** Duplicate thick and thin blood films were prepared from the capillary blood of each study participant and examined for species identification and parasite density on day zero to confirm adherence to the inclusion criteria and at each follow-up day. A fresh 10% Giemsa stain was prepared at least once a day. Thick and thin blood films were stained with 10% Giemsa for 10 minutes and examined by two well-trained qualified microscopists. Parasite densities were calculated by averaging the two counts. If there was discordant, re-examined by a third independent microscopist, and parasite density was calculated by averaging the two closest counts. Parasitemia was calculated by counting the number of asexual parasites against 200 white blood cells (WBCs) and then multiplying by an assumed average white blood cell density (8000 per μl) as listed below [14].

$$\text{Parasite density (per μl)} = \frac{\text{number of parasites count} \times 8000}{\text{Number of leukocytes counted}}$$

**Hemoglobin examination.** Determination of hemoglobin concentration was done by Hemo Cue HB 301+ analyzer (Hemo Cue, Angelholm, Sweden) from peripheral blood collected via finger pricking using sterile disposable lancets on the day, 0, 14, and 28. The photometric determination of hemoglobin was performed by the entry of a drop of blood to the optical window of the micro cuvette placed into the cuvette holder and absorbance was measured spectrophotometrically at 540 nm. Classification of anemia was based on hemoglobin cut-off values set by the WHO (Hgb 7–9.9 g/dl for <5, and 8–10.9 g/dl moderate anemic for 5–15, and >15 of age respectively. Hgb 10–10.9 g/dl, 11–11.9g/dl and 11–12.9g/dl mild anaemic for <5, 5–15 and non-pregnant women and adult male respectively, and Hgb >11g/dl for <5, >11.5g/dl for 5–15 and non-pregnant women and >13g/dl for adult male respectively classified as non anemic) [15].

## Study endpoint

The study end-point is assigned to a patient based on the WHO definition of treatment outcome; valid study end-points include treatment failure during the study period early treatment failure (ETF) danger sign or severe malaria on day 1, 2, or 3 in the presence of parasitemia; parasitemia on day 2 higher than on day 0, irrespective of axillary temperature; parasitemia on day 3 with axillary temperature ≥37.5 ˚C and parasitemia on day 3 ≥ 25% of count on day 0. patients having danger signs or symptoms, severe malaria in the presence of parasitemia on any day between days 4 and 28 in patients who did not previously meet any of the criteria of

early treatment failure; the presence of parasitemia on any day between days 4 and 28 with axillary temperature ≥37.5 ˚C in patients who did not meet any of the criteria of early treatment failure were classified under late clinical failure (LCF). Presence of parasitemia on any day between day 7 and 28 and axillary temperature ≤ 37.5 ˚C in patients who did not previously meet any of the criteria of early treatment failure or let clinical failure were categorized under late parasitological failure (LPF). Those patients who completed the follow-up period without treatment failure were classified under adequate clinical and parasitological response (ACPR) and patients who failed to return to the study visit were assigned to lost to follow-up (LFU) [15].

## Safety and quality

Safety and adverse events were assessed by recording the nature and incidence of any events following treatment. Adverse events are defined as unfavorable intended sign symptoms not presented at baseline but occurring during follow-up or manifested at baseline and increased in intensity during follow-up or worsen with the use of CQ. A severe adverse effect is defined by WHO as, any untoward medical occurrence that any dose results in death, life-threatening, and requires hospitalization or prolongation of hospitalization [15]. Based on the stated criteria adverse events were classified and reported to Ethiopian Public Health Institute as required. The data was collected accordingly to the protocol. Standard operating procedures are implemented in each laboratory activity. Quality of the reagent and equipment was maintained as per the standard operating procedures.

## Statistical analysis

All data were double entered into the WHO Excel sheet which is designed for analysis of therapeutic efficacy study data. Data was also entered into Statistical Package for Social Science version 21 (SPSS 21.0) software. One-Way ANOVA and independent sample t-tests were used to compare baseline temperature and parasitemia between age groups, and paired sample t-test was used to compare mean Hgb levels between D0 and D14, D0 and D28, D14 and D28. The risk of therapeutic failure and cure rates were estimated using per-protocol and K-M survival (censored) analysis methods. All comparisons were performed at 95% CI and a significance level of 0.05.

## Ethics approval and consent to participate

Ethical clearance was obtained from the Ethical Clearance Committee of Bahir Dar University Institutional Review Board, Ethiopia (P162/2021), and Ethiopian Public Health Institute (EPHI) (P294/2020). Permission letter was obtained from Amhara Public Health Institute (APHI) before the study commenced. Additional permission was obtained from Shewa Robit Health Centre. Written informed consent was obtained from adult patients while for children assent was obtained from their parents or guardians.

## Results

### Characteristics of study participants

A total of 2090 malaria suspected patients were screened, females accounted for 54.1%. About 8% (167/2090) of the participants were found to be slide positive for malaria, of which 147 (88%) of the cases were attributed to *P. vivax*. Ninety (61%) of the *P. vivax* mono-infection cases that fulfilled the inclusion criteria were recruited. The majority of the study participants 83 (92.2%) were urban residents.

Among the 90 enrolled participants, males took the highest number yielding a male-to-female ratio of about 2 (58/32). The median age of the study participants was 18 years, ranging from 1.4–60 years, and under-five children accounted for 11.1%. About 50 (55.6%) of the participants had access to a bed net, with a proper bed net utilization rate of 56%. Among 78 (86.8%) of the participants who had a previous history of malaria attack, three-quarters took chloroquine and the remaining 11 (12.2%) were treated with Arthemeter Lumefantrine. Of the enrolled participants, four participants couldn't complete the 28-day follow-up and were therefore excluded (Fig 1).

About 91.1% were febrile at the time of enrolment and the remaining had a history of fever in the previous 48 hrs. The mean baseline body temperature (±) SD was 38.8±1˚C (males 38.8

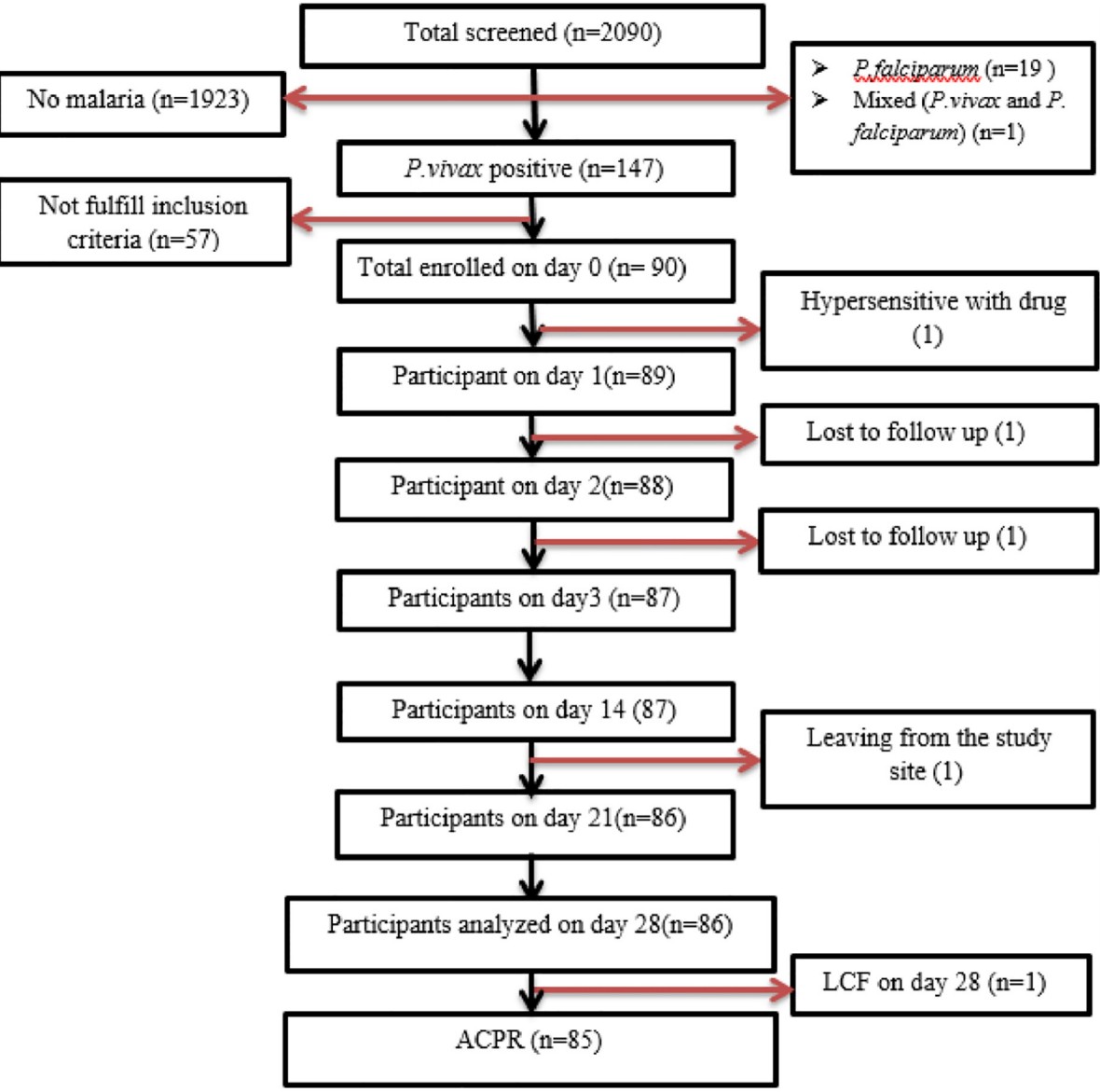

**Fig 1. Flow chart of study participants' recruitment and follow-up at Shewa Robit Health Centre, Northeast Ethiopia from November 2020 to March 2021.**

**Table 1. Baseline characteristics of the study participant with chloroquine treatment in Shewa Robit Health Centre, Northeast Ethiopia from November 2020 to March 2021.**

| Variables | | Sex | | Residence | | Total |
|---|---|---|---|---|---|---|
| | | Male | Female | Urban | Rural | |
| **No (%)** | | 58 (64.4) | 32 (35.6) | 83 (92.2) | 7 (7.8) | 90 (100) |
| **Mean Temp (˚C)** | | 38.8 | 39 | 38.8 | 39.4 | 38.8 |
| **Mean Hgb(g/dl)** | | 13.4 | 12.4 | 13 | 14 | 13 |
| **Anemia status** | Mild n(%) | 8 (13.8) | 9 (28.1) | 15 (18.1) | 2(28.6) | 17(18.9) |
| | Moderate n(%) | 4 (6.9) | 3 (9.4) | 7 (8.4) | 0 (0) | 7 (7.8) |
| | Total n(%) | 12 (20.7) | 12 (37.5) | 22 (26.5) | 2 (28.6) | 24 (26.7) |
| **Mean (Geo) Para/μl** | | 5097 | 5895 | 5261 | 6810 | 5368 |
| **Gametocyte carriage n (%)** | | 56 (96.6) | 31(96.9) | 80 (96.4) | 7 (100) | 87 (96.7) |
| **<10000 parasitemia n (%)** | | 45 (77.6) | 22 (68.8) | 62 (74.7) | 5 (71.4) | 67 (74.4) |
| **>10000 parasitemia n (%)** | | 13 (22.4) | 10 (31.2) | 21 (25.3) | 2 (28.6) | 23 (25.6) |

Geo = Geometric, Hgb = Haemoglobin, Para = Parasitaemia, Temp = Temperature.

±0.9 ˚C, females 39±1.2 ˚C) with no significant difference (*P = 0.427*). The highest mean body temperature was recorded for under-five children (39.3±0.1 ˚C). Headache, vomiting, and nausea were the major clinical signs/symptoms reported on the first day (D0). Headache was the most common symptom 69 (80.2%). The average weight and height were 41.1kg and 145.7cm, respectively.

The baseline mean parasitemia was 5368 (geometric mean) with significant variation among age groups (p = 0.001). Overall gametocyte carriage at baseline was 87/90 (96.7%). The baseline means hemoglobin level of the study participants was 13 g/dl, with 11.5, 11.9, and 13.8 g/dl for <5, 5–15, and >15 age groups, respectively. The prevalence of anemia was 26.7% (18.9% mild and 7.8% moderately anemic) (Table 1).

Parasite density showed relatively decreasing numbers with age. Baseline parasite load was higher in children compared to adults (day of admission) (r = -0.340, $r^2 = 0.115$, significant at *P = 0.001*) (Fig 2).

## The cure rate of chloroquine

The overall cure rate of chloroquine was 98.8% (95% CI: 95.3–100%). On the 28th day of follow-up, one patient was identified to be infected by *P.vivax* and classified as late clinical failure (LCF) 1.2% (95%CI: 0.0–4.7%) (PCR not corrected) (Table 2).

The K-M survival analysis over the 28-day follow-up period showed a cumulative success rate of 98.8% (95% CI: 95.3–100%) (Table 3, Fig 3).

## Parasite clearance

All asexual stages and gametocytes were cleared within 48 hrs (day 2 of the follow-up period). A 1.5 years old age female child had a late clinical failure with a parasitemia of 1760/μl of blood on day 28. However, the parasitemia level of the patient with treatment failure on the 28th day of infection (1760/μl) was lower than the day of admission (4560/μl), giving a parasite reduction ratio (PRR) of 2.6/μl (Fig 4).

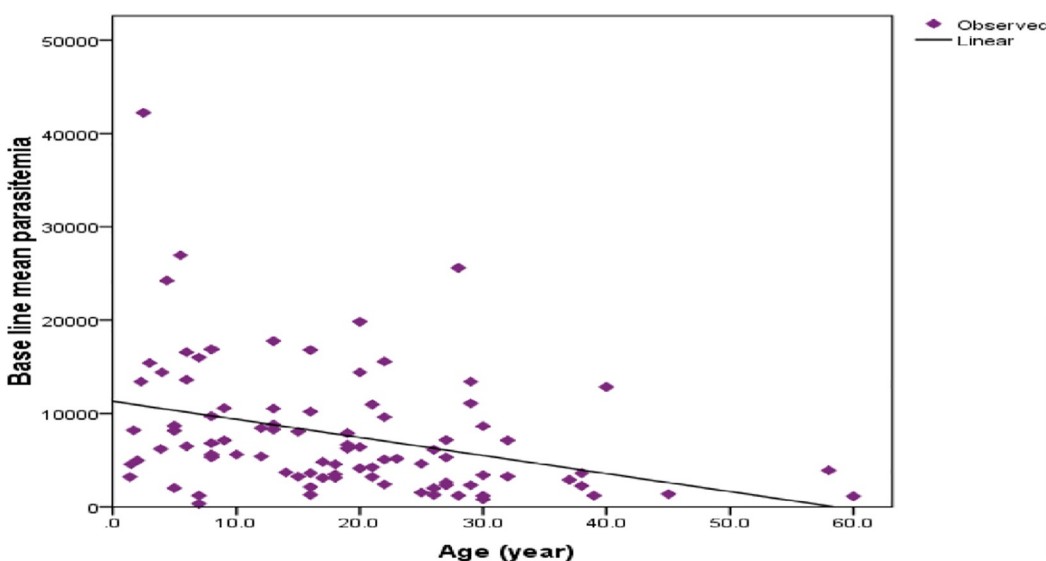

**Fig 2. Relation between age and parasite counted at baseline of study participants in Shewa Robit Health Centre, northeast Ethiopia from November 2020 to March 2021.**

**Table 2. Per protocol treatment outcome chloroquine at day 28, Shewa Robit Health Center, northeast Ethiopia from November 2020 to March 2021.**

| Outcome | <5 | 5–15 | >15 | Sex | | Total | 95%CI |
|---|---|---|---|---|---|---|---|
| | | | | Male | Female | | |
| ETF n (%) | 0 (0) | 0 (0) | 0 (0) | 0 (0) | 0 (0) | 0 (0) | |
| LCF n (%) | 1 (10) | 0 (0) | 0 (0) | 0 (0) | 1 (3.2) | 1(1.2) | 0.00–4.7% |
| LPF n (%) | 0 (0) | 0 (0) | 0 (0) | 0 (0) | 0 (0) | 0 (0) | |
| ACPR n (%) | 9 (90) | 25 (100) | 51 (100) | 55 (100) | 30 (96.8) | 85 (98.8) | 95.3–100% |
| Total analyzed n (%) | 10 (100) | 25 (100) | 51(100) | 55 (100) | 31(100) | 86 (100) | |

## Fever clearance

Of the 86 participants that completed the study 79 (91.9%) patients had a body temperature of 37°C or above and the rest had a history of fever in the last 48hrs. However, 57 (66.3%) participants cleared fever on days 1, 74 (86%), and 78 (90.7%) on days 2 and 3 respectively. Nearly all participants cleared their fever on day 7 (Fig 5).

**Table 3. Chloroquine treatment outcome of study participants based on K-M analysis in Shewa Robit Health Center, northeast Ethiopia from November 2020 to March 2021.**

| Follow up days | At-Risk | Censored | Failure | Survived | K-M survival rate | K-M Failure rate |
|---|---|---|---|---|---|---|
| 0 | 90 | 1 | 0 | 90 | 1 | 0 |
| 1 | 89 | 1 | 0 | 89 | 1 | 0 |
| 2 | 88 | 1 | 0 | 88 | 1 | 0 |
| 3 | 87 | 0 | 0 | 87 | 1 | 0 |
| 7 | 87 | 0 | 0 | 87 | 1 | 0 |
| 14 | 87 | 1 | 0 | 87 | 1 | 0 |
| 21 | 86 | 0 | 0 | 86 | 1 | 0 |
| 28 | 86 | 0 | 1 | 85 | 0.988372 | 0.011628 |

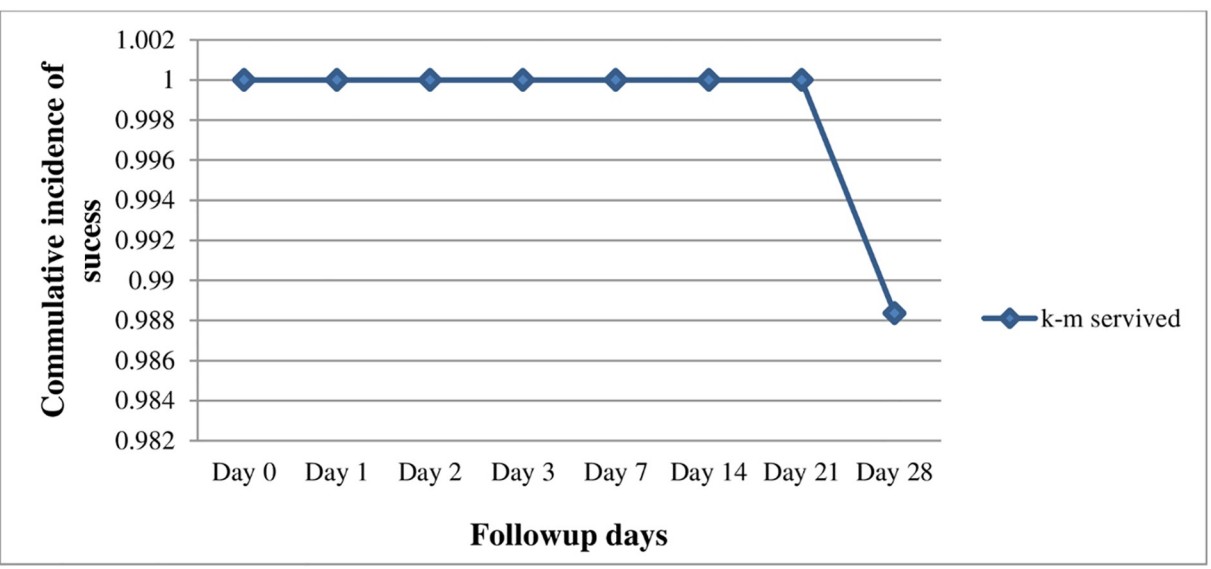

**Fig 3. Kaplan-Meier survival curve of the study outcome in Shewa Robit Health Center, northeast Ethiopia from November 2020 to March 2021.**

## Hemoglobin determination

Significant recovery of hemoglobin occurred on follow-up days. On the day of recruitment (day 0), about 19.8% and 8.1% of the participants were mildly and moderately anemic, respectively. Of 24 mild and moderate anemic patients, only 12.5% had no previous history of malaria and fully recovered on day 28. The remaining 87.5% of anemic patients had a history of repeated malaria infection and 81% of them recovered; the rest 19% of patients couldn't

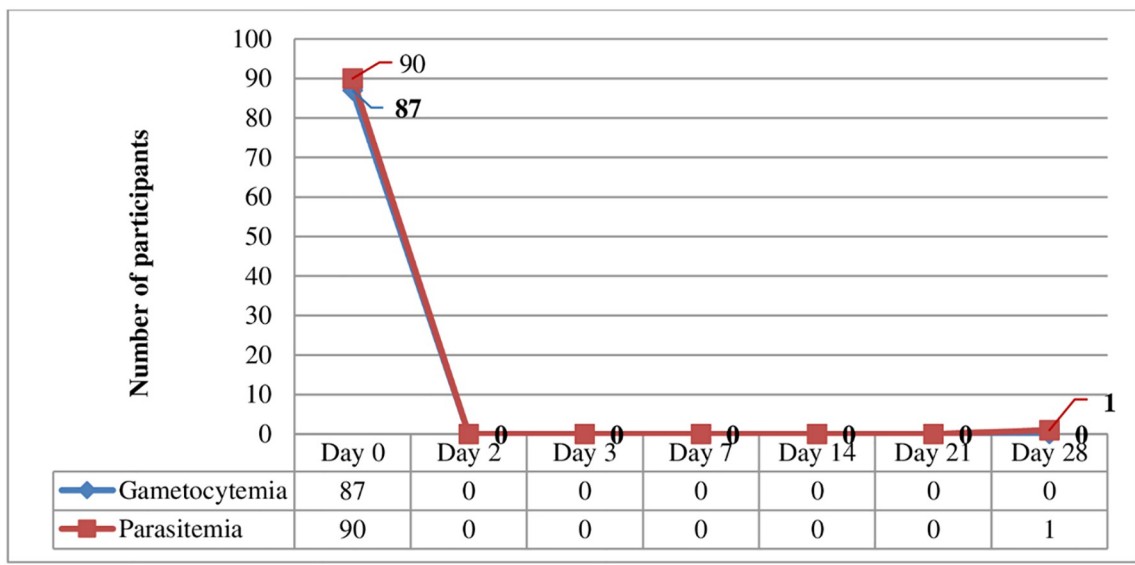

**Fig 4. A pattern of parasite and gametocyte clearance following chloroquine treatment in Shewa Robit Health Centre, northeast Ethiopia from November 2020 to March 2021.**

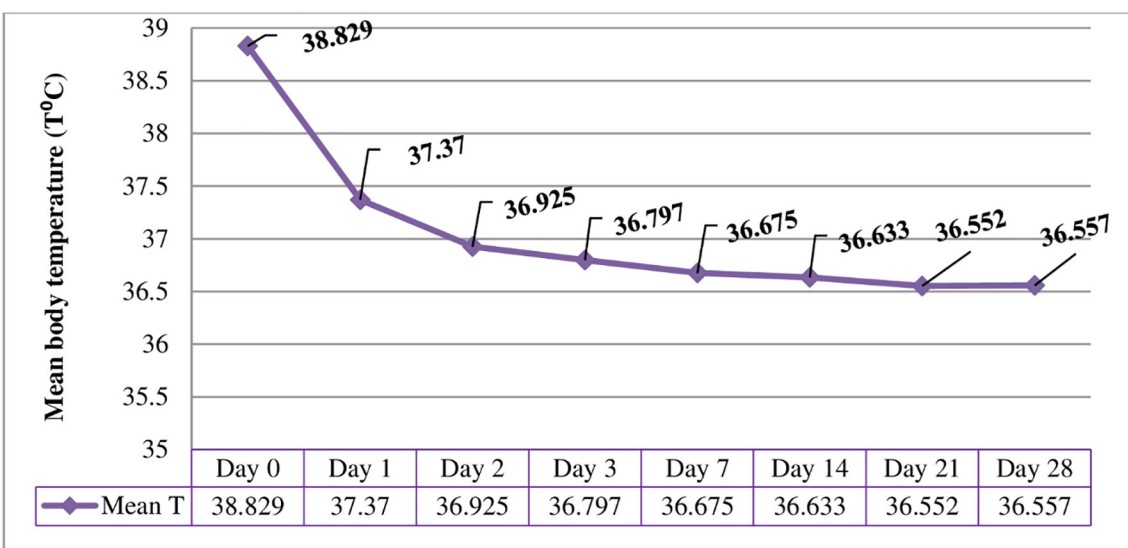

**Fig 5. Mean axillary temperature on days zero to 28 among chloroquine treated *P. vivax* infected participants in Shewa Robit Health Center, northeast Ethiopia from November 2020 to March 2021.**

show improvement in hemoglobin level even on day 28. The number of anemic patients decreased from day zero before CQ treatment 24 (27.9%) to 14 (16.3%) and 4 (4.7%) on days 14 and 28 respectively following CQ therapy (Table 4).

## Adverse events following treatment

One 19-year-old male patient developed hypersensitive to the drug and exhibited recurrent vomiting, fatigue, and weakness 4 hrs after taking CQ, and was therefore excluded from the study and treated with the second-line drug accordingly to the national malaria diagnosis and treatment protocol. At baseline fever, headache, and vomiting were the most encountered signs/symptoms, accounting for 80.2%, and 36%, respectively. Adverse events observed following chloroquine treatment were mouth ulcers (11.6%) and blurred vision (2.3%). The number of patients with abdominal pain and cough also showed an increment from the baseline (1.2% vs 9.3% & 2.3% vs 11.6%, respectively). However, all clinical symptoms and adverse events declined on day 7 and thereafter (Table 5).

## Discussion

Antimalarial drug resistance has been a major concern in malaria prevention, control, and management. In Ethiopia, CQ has been extensively used as a first-line drug for the treatment of uncomplicated *P.vivax*. However, some recent data showed that CQ resistance has been reported in different parts of the country [10,16], and elsewhere [17].

The current study showed high therapeutic efficacy of chloroquine at 28 days of follow-up 98.8% (95% CI: 95.3%-100%). It is consistent with findings from different parts of Ethiopia; Hossana, 96.7% [11], Metekel, 97.3% [18], Jimma, 97.3% [12], Wolkite, 97.5% [19] and elsewhere 99.5% [20].

Contrary to the current finding, higher treatment failure of CQ was reported on different follow-up days in different parts of Ethiopia including from the current study site 6.6% (7 years ago) [21], Halaba district, 13%, [16] Debrezeit and Nazareth towns, 7.5% [22], Southern

**Table 4. Anemia status among study participants following chloroquine treatment in Shewa Robit Health Center, northeast Ethiopia from November 2020 to March 2021.**

| Variables | | | Follow up days | | |
|---|---|---|---|---|---|
| Age group | | | Day 0 | Day14 | Day 28 |
| | <5 (n = 10) | Mean Hgb level | 11.5 | 11.8 | 12.4 |
| | | Mild n (%) | 2 (20) | 2 (20) | 0 (0) |
| | | Moderate n (%) | 1 (10) | 1 (10) | 2 (20) |
| | 5–15 (n = 25) | Mean Hgb level | 11.9 | 12.6 | 13.1 |
| | | Mild n (%) | 6 (24) | 1 (4) | 0 (0) |
| | | Moderate n (%) | 3 (12) | 2(8) | 0 (0) |
| | >15 (n = 51) | Mean Hgb level | 13.8 | 13.9 | 14.3 |
| | | Mild n (%) | 9 (17.6) | 7 (13.7) | 1 (2) |
| | | Moderate n (%) | 3 (5.9) | 1 (2) | 1 (2) |
| Sex | Male (n = 55) | Mean Hgb level | 13.3 | 13.6 | 14 |
| | | Mild n (%) | 8 (14.5) | 7 (12.7) | 1 (1.8) |
| | | Moderate n (%) | 4 (7.3) | 1 (1.8) | 1 (1.8) |
| | Female (n = 31) | Mean Hgb level | 12.3 | 12.8 | 13 |
| | | Mild n (%) | 9 (29) | 3 (10) | 0 (0) |
| | | Moderate n (%) | 3 (10) | 3 (10) | 2 (6.5) |
| Total | | Mean Hgb (g/dl) | 13 | 13.3 | 13.7 |
| | | Mild n (%) | 17 (19.8) | 10 (11.6) | 1 (1.2) |
| | | Moderate n (%) | 7 (8.1) | 4 (4.7) | 3 (3.5) |
| | | All anemic n (%) | 24 (27.9) | 14 (16.3) | 4 (4.7) |
| Mean difference | | | Day 0 and 14 | Day 0 and 28 | Day14 and 28 |
| | | | P = 0.032 | P<0.001 | P = 0.005 |

Ethiopia, 9.4% [8], and elsewhere Madagascar 5.1% [23]. Relapse, malabsorption of drug, reinfection and other patient/drug-related factors could be contributing factors to treatment failure of *P.vivax* [24]. Complementary laboratory tests such as the measurement of CQ and Desethylchloroquine (DCQ) concentration levels in the blood and genotyping would be helpful to distinguish between reinfection and recrudescence.

**Table 5. Common malaria clinical signs and symptoms and adverse events following Chloroquine treatment in Shewa Robit Health Centre, northeast Ethiopia from November 2020 to March 2021.**

| Adverse events/clinical symptoms | Follow up days | | | | | | | |
|---|---|---|---|---|---|---|---|---|
| | Day 0 | Day 1 | Day 2 | Day 3 | Day 7 | Day 14 | Day 21 | Day 28 |
| Headache n (%) | 69 (80.2) | 41 (47.7) | 20 (23.3) | 9 (10.5) | 4 (4.7) | 0 (0) | 0 (0) | 0 (0) |
| Anorexia n (%) | 8 (9.3) | 6 (7) | 6 (7) | 3 (3.5) | 0 (0) | 0 (0) | 0 (0) | 0 (0) |
| Nausea n (%) | 19 (22.1) | 5 (5.8) | 4 (4.7) | 0 (0) | 0 (0) | 0 (0) | 0 (0) | 0 (0) |
| Vomiting n (%) | 31 (36) | 15 (17.4) | 3 (3.5) | 0 (0) | 0 (0) | 0 (0) | 0 (0) | 0 (0) |
| Abdominal pain n (%) | 1 (1.2) | 2 (2.3) | 5 (5.8) | 1 (1.2) | 0 (0) | 0 (0) | 0 (0) | 0 (0) |
| Diarrhoea n (%) | 3 (3.5) | 0 (0) | 1 (1.2) | 1 (1.2) | 1 (1.2) | 0 (0) | 0 (0) | 0 (0) |
| Cough n (%) | 2 (2.3) | 3 (3.5) | 4 (4.7) | 3 (3.5) | 0 (0) | 0 (0) | 0 (0) | 0 (0) |
| Behavioral change n (%) | 6 (7) | 1 (1.2) | 1 (1.2) | 0 (0) | 0 (0) | 0 (0) | 0 (0) | 0 (0) |
| Dizziness n (%) | 2 (2.3) | 1 (1.2) | 2 (2.3) | 3 (3.5) | 0 (0) | 0 (0) | 0 (0) | 0 (0) |
| Mouth ulcer n (%) | 0 (0) | 2 (2.3) | 3 (3.5) | 4 (4.5) | 1(10.2) | 0 (0) | 0 (0) | 0 (0) |
| Blurred vision n (%) | 0(0) | 0 (0) | 1 (1.2) | 1 (1.2) | 0 (0) | 0 (0) | 0 (0) | 0 (0) |

Treatment failure was observed at day 28, 1.2% (95% CI: 0.0%-4.7%) in a 1.5-year-old female child. This is in agreement with earlier studies in the country [18,21,25] and Madagascar [23], in which the majority of treatment failures were observed in children, However, previous studies in Ethiopia couldn't demonstrate a significant association of age with treatment failure [12,19]. The parasitemia load on the day of failure was lower (1760/μl) than on the day of admission (4560/μl). The parasite reduction ratio (PRR) of the treatment failure cases was 2.6/μl, which is similar to the study carried out in different areas of Ethiopia [16,19,21].

There was a rapid clearance of parasite density in the study participants over two days of CQ administration. This could be the effectiveness of CQ in curing *P.vivax* [26] and could be a rapid and complete absorption of CQ in the gastrointestinal tract following oral administration [27].

Fever clearance time is crucial for evaluating the therapeutic effectiveness of chloroquine in the treatment of *P.vivax*, as fever is one of the criteria for monitoring and classifying responses to CQ treatment [14]. In the present study, 91.9% of patients had fever at baseline, as malaria fever is elevated due to the cyclical release of merozoite and malaria toxins during schizont rupture of red blood cells, which induce endogenous pyrogens [28]. After CQ treatment, more than 90% of the participants cleared fever within three days follow-up period, which is in line with studies conducted in the country [29]. Rapid clearance of fever could be due to the antipyrogenic effect of CQ [30] and/or inhibition of malaria hemozoin crystal formations [31].

Age and parasitemia were negatively correlated in our study; parasite load was higher in children than in adults. These findings are similar to the previous study in Serbo [16], and Shewa Robit [21]. This might be due to acquired immunity developed through previous exposure to *P.vivax* infection [26]. The mean hemoglobin level of the study participants was improved significantly, similar to the study conducted in Wolkite [19] and elsewhere12.5g/dl to 13.2g/dl [20].

Destruction of infected red blood cells (RBCs) and removal of a high number of uninfected RBCs leads to malaria-related anemia [32]. In the present study, there was no severe anemia recorded, which is comparable to the finding in Wolkite [19]. The possible mechanism involved in severe malaria anemia is a cumulative loss of RBCs due to mixed infection, lysis of uninfected RBCs in the circulation, and impaired RBC production [33,34]. The majority of mild and moderate anemic patients recovered on day 28; the rest few anemic patients who had a history of repeated malaria infection couldn't show improvement in hemoglobin level even on day 28. Anemia can be linked to other factors in addition to malaria infections such as nutritional deficiencies and worm infections [35–37]. Repeated malaria infection and relapse can cause impairment of hemoglobin levels [38].

Chloroquine can cause side effects or intensify malaria symptoms already present such as fever, headache, anorexia, nausea, vomiting, abdominal pain, diarrhea, cough, behavioral change, and dizziness [39]. In this study, most of the observed adverse events were similar to the common symptoms of malaria mentioned above and disappeared following treatment within seven days. But the frequency of abdominal pain and cough increase following CQ treatment. A similar result was reported in Brazil [40]. Mouth ulcers and blurred visions were also observed after CQ treatment in this study. However, the proportion of blurred vision observed in this finding is much lower, 2/86 (2.3%) as compared to 27/50 (54%) patients from Brazil [40].

Although this study showed a good treatment response, other supplementary techniques that we did not include in our study; such as determining blood CQ level and molecular techniques in classifying the treatment failures would further strengthen the study outcomes. A wide-ranging geographic coverage can also improve the reliability of QC status in the treatment of uncomplicated *P.vivax*.

## Conclusion

The present study revealed the cure rate of CQ remains high for the treatment of uncomplicated *P.vivax* infection with rapid clearance of parasitemia and fever; hemoglobin improvement and good clinical resolution. Severe adverse event was not recorded during the follow-up period. Our study result complements the current use of CQ for the treatment of uncomplicated *P. vivax* by the national malaria elimination program. Given the contradictory reports, continuous, and strong surveillance of therapeutic efficacy studies on CQ is needed for early detection and effective control of measures on the possible emergence and spread of drug resistance in the study area and the country at large.

## Supporting information

**S1 Raw data.**
(SAV)

**S1 File.**
(DOCX)

## Acknowledgments

We would like to thank Ethiopian Public Health Institute (EPHI), Amhara regional health bureau and Shewa Robit health center and Bahir Dar University for their unlimited support and expertise input for this project. We would also like to acknowledge the ministry of health through EPHI for financial and technical support. The study drugs were obtained through the WHO Ethiopia office. We also acknowledge the study team members and study participants.

## Author Contributions

**Conceptualization:** Habtamu Belay.

**Formal analysis:** Habtamu Belay.

**Funding acquisition:** Mebrahtom Haile, Gudissa Assefa.

**Investigation:** Habtamu Belay.

**Methodology:** Habtamu Belay, Hussein Mohammed, Heven Sime.

**Project administration:** Ashenafi Assefa.

**Resources:** Hussein Mohammed, Heven Sime, Henok Hailegeorgies, Bokretsion Gidey, Worku Bekele, Ashenafi Assefa.

**Software:** Habtamu Belay, Megbaru Alemu, Mihreteab Alebachew Reta, Andargachew Almaw Tamene.

**Supervision:** Heven Sime, Henok Hailegeorgies, Bokretsion Gidey, Geremew Tasew, Ashenafi Assefa.

**Visualization:** Megbaru Alemu, Tadesse Hailu, Ashenafi Assefa.

**Writing – original draft:** Habtamu Belay.

**Writing – review & editing:** Megbaru Alemu, Tadesse Hailu, Ashenafi Assefa.

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
