## [Decision Letter · Decision Letter 0]

26 Jul 2022

PONE-D-22-18161Therapeutic efficacy of Chloroquine for the treatment of uncomplicated Plasmodium vivax infection in Shewa Robit, Northeast EthiopiaPLOS ONE

Dear Dr. Temesgne

Thank you for submitting your manuscript to PLOS ONE. After careful consideration, we feel that it has merit but does not fully meet PLOS ONE’s publication criteria as it currently stands. Therefore, we invite you to submit a revised version of the manuscript that addresses the points raised during the review process.

ACADEMIC EDITOR: The maniscript is interesting, but several issue related to the methodology section, inclusion and exclusion creiteria and sample effort should be adressed in the manuscript. Please follow the reccomendations of the reviewers.

We look forward to receiving your revised manuscript.

Kind regards,

José Luiz Fernandes Vieira

Academic Editor

PLOS ONE

Journal Requirements:

"no the funders has no role in study design, data collection and analysis, decision to publish, or preparation of the manuscript."

Reviewers' comments:

Reviewer's Responses to Questions

**Comments to the Author**

1. Is the manuscript technically sound, and do the data support the conclusions?

Reviewer #1: Yes

Reviewer #2: Partly

2. Has the statistical analysis been performed appropriately and rigorously? 

Reviewer #1: Yes

Reviewer #2: I Don't Know

3. Have the authors made all data underlying the findings in their manuscript fully available?

Reviewer #1: Yes

Reviewer #2: Yes

4. Is the manuscript presented in an intelligible fashion and written in standard English?

Reviewer #1: Yes

Reviewer #2: Yes

5. Review Comments to the Author

Reviewer #1: Dear,

Congratulations on the study, it has great scientific relevance. Therefore, some

questions need to be answered:

When analyzing your study, both the abstract and the introduction are contemplating

the necessary information; The methodology needs to be further explored. It is still not

entirely clear what the inclusion and exclusion criteria for the recruited patients were.

In addition, the sample number seems small to me. Wouldn't it have a chance to

increase? Another important factor is that no analysis of the biochemical parameters of

the patients was performed, these data would give you significant answers to your

findings. Why wasn't it done? The results are well described, but at the same time I

noticed that the patients did not present itching. Typically, patients who use CQ have

these reactions. Was it not identified or did these patients have and/or abandoned the

segment? Furthermore, I suggest that in figures 3 and 4 you put the corresponding

values in each highlight point for a better understanding for the reader; The discussion

presented includes the necessary information. However, by addressing minor

limitations you could emphasize that the process of genetic polymorphism could

account for drug resistance. This is constantly being discussed in several scientific

articles in different locations worldwide. It is also necessary for you to present more

accurate information to justify your hypotheses and/or findings; The conclusion is ok.

Also, what makes your work different?

I hope that this information has been essential for a better understanding of the study.

Thank you very much in advance.

Reviewer #2: Background

1. Put epidemiology data latest

2. When did the first case of resistance appear in Ethiopia? Insert this information

3. How was resistance studies carried out previously carried out?

Methods

4. describe what were the inclusion criteria

5. describe what were the exclusion criteria?

6 describe what were the severe malaria exclusion criteria?

7. Where did this frequency of therapeutic failure come from?

8. What is the expected minimum and maximum percentage for sample calculation?

9. what number of malaria in the studied place for sample calculation? need your population number.

10. what definition of recurrence and clearance of parasitemia?

11. Drug quality analysis was performed? Enter chloroquine information?

12. Did all patients take the same batch of chloroquine?

13. Was it quality control carried out on this batch?

14. Why were these follow-up visits selected? if the patient had a return of malaria on other days or after D28 was it lost?

15. How was treatment on different days describe dosage on each day?

16. Were recorded Baseline data on socio-demographic and clinical characteristics?

17. Has quality assurance been performed for microscopy?

18. What were the criteria for establishing adverse effects?

19. How was security assessed? what criteria?

20. Primary Outcomes and secondary? Define each item of this study.

21. Was it not evaluated clearance and presence of gametocytes?

Results

22. Make a flowchart describing how the selection was made and the number of participants included.

23. Were participants who did not take CQ as treatment included?

24. This phrase "couldn't complete the 28-day follow-up and were therefore excluded", needs to be included in the flowchart.

25. In figure 2, insert the correlation confidence interval? p value?

26. what purpose of making correlation?

27. Why was the dosage of chloroquine not performed for classification of recurrence?

28. Why were these data (K-M analysis) not presented through the survival curve?

29. what was the parasitemia of the recurrent case? Put in the figure 3.

30. This study cannot speak about resistance to CQ, as resistance was not confirmed in these participants.

31. What was done to confirm resistance?

Discussion

32. Relapse, malabsorption of drug, poor drug quality and re-infection could be contributing factors to treatment failure of P. vivax? Insert into the discussion on this aspect.

33. drug quality control was not performed?

34. Better describe all limitations: sample size, not having performed chloroquine dosage, not performing drug quality control

6. PLOS authors have the option to publish the peer review history of their article (what does this mean?). If published, this will include your full peer review and any attached files.

Reviewer #1: No

Reviewer #2: No

---

## [Author Response · Author response to Decision Letter 0]

16 Sep 2022

Response to reviewers

1. Point by point response to reviewer #1

Comments to the Author

1. Is the manuscript technically sound, and do the data support the conclusions?

Reviewer #1: Yes

2. Has the statistical analysis been performed appropriately and rigorously? 

Reviewer #1: Yes

3. Have the authors made all data underlying the findings in their manuscript fully available?

Reviewer #1: Yes

4. Is the manuscript presented in an intelligible fashion and written in standard English?

Reviewer #1: Yes

5.Review Comments to the Author

Reviewer #1: Dear, Congratulations on the study, it has great scientific relevance. Therefore, some questions need to be answered: When analyzing your study, both the abstract and the introduction are contemplating the necessary information; The methodology needs to be further explored. It is still not entirely clear what the inclusion and exclusion criteria for the recruited patients were. In addition, the sample number seems small to me. Wouldn't it have a chance to

increase? Another important factor is that no analysis of the biochemical parameters of

the patients was performed, these data would give you significant answers to your

findings. Why wasn't it done? The results are well described, but at the same time I

noticed that the patients did not present itching. Typically, patients who use CQ have

these reactions. Was it not identified or did these patients have and/or abandoned the

segment? Furthermore, I suggest that in figures 3 and 4 you put the corresponding

values in each highlight point for a better understanding for the reader; The discussion

presented includes the necessary information. However, by addressing minor

limitations you could emphasize that the process of genetic polymorphism could

account for drug resistance. This is constantly being discussed in several scientific

articles in different locations worldwide. It is also necessary for you to present more

accurate information to justify your hypotheses and/or findings; The conclusion is ok.

Also, what makes your work different? I hope that this information has been essential for a better understanding of the study. Thank you very much in advance.

Response: Thank you for your constructive comments, we accepted all your suggestions and tried to correct them accordingly. 

1. We have followed strictly the WHO guideline “Methods for the surveillance of antimalarial drug efficacy (2009)” accordingly we have followed strict inclusion and exclusion criteria as well as sample size calculation and corrected accordingly in the revised manuscript. The 73-size is the minimum required for such a study.

2. We strongly agree that additional biomedical analysis may strengthen the study however due to strictly following the guideline and limited resources, we were not able to do the biomedical analysis.

3. Itching can be manifested in patients taking CQ. Unfortunately, we observed participants physically and didn’t see any itchy behavior. The participants may not have noticed and didn’t explain it to us.

4. Thank you, comments on figure 3 & 4 corrected accordingly on the revised manuscript.

5. We agree that genetic polymorphism may contribute to drug resistance however, it was beyond the scope of this study, and we have indicated it as a limitation thank you for raising it.

6. Our study may not be totally unique but we produce evidence from a common location and the information is relevant for local and global malaria control efforts. 

Point by point response to reviewer#2

Comments to the Author

1. Is the manuscript technically sound, and do the data support the conclusions?

Reviewer #2: Partly

2. Has the statistical analysis been performed appropriately and rigorously? 

Reviewer #2: I Don't Know

3. Have the authors made all data underlying the findings in their manuscript fully available?

Reviewer #2: Yes

4. Is the manuscript presented in an intelligible fashion and written in standard English?

Reviewer #2: Yes

5. Review Comments to the Author

Reviewer#2: Background

1. Put epidemiology data latest

Response: thank you for your suggestion. We have updated the epidemiology data.

2. When did the first case of resistance appear in Ethiopia? Insert this informatioitn

Response: thank you, we have corrected accordingly.

3. How was resistance studies carried out previously carried out?

Response: Thank you, WHO recommends routine monitoring of drug resistance using its guideline every other year for two to six sites based on country size. There have been continuous but inconsistent studies of malaria drug resistance for the past three decades in Ethiopia. 

Methods

4. describe what were the inclusion criteria

Response: Thank you, corrected accordingly. 

5. describe what were the exclusion criteria?

Response: Thank you and accepted.

6 describe what were the severe malaria exclusion criteria?

Response: Thank you. The severe malaria exclusion criteria are described in the WHO study protocol that we followed. We want to minimize the size of the MS and keep it as a reference.

7. Where did this frequency of therapeutic failure come from?

Response: thank you for your question, the frequency of failure was obtained from a previously published study. 

8. What is the expected minimum and maximum percentage for sample calculation?

Response: Thank you. According to WHO guidelines, in the TES study, the sample size is characterized and calculated to be the minimum number of individuals required for an expected 5% failure is 73, and a maximum of >88 patients.

9. what number of malaria in the studied place for sample calculation? need your population number.

Response: Thank you. The population may be important but all self-presenting patients suspected of malaria were diagnosed and if P.vivax infection is confirmed with the parasitemia load, were enrolled in the study team. The study population is the catchment area for Shewa Robit HC, it is estimated to be 60,234.

10. what definition of recurrence and clearance of parasitemia?

Response: we have appreciated your question, but the terms clearance and recurrence are well stated in the study protocol that clearance is parasitemia disappearing from the patient sample when examined by microscopy. Recurrence is the reappearance of the parasite in the patient’s blood during the follow-up period.

11. Drug quality analysis was performed? Enter chloroquine information?

Response: Thank you. The quality of the study drug was checked and provided by WHO. Information (drugs’ name, manufacturer, lot number, and expired date) are described in the manuscript. No additional quality assessment has been made.

12. Did all patients take the same batch of chloroquine?

Response: Thank you. All drugs were in one batch.

13.Was it quality control carried out on this batch?

Response: Thank you. The quality concern was done by WHO, as drug provided by.

14. Why were these follow-up visits selected? if the patient had a return of malaria on other days or after D28 was it lost?

Response: Thank you, follow-up visits were scheduled based on WHO therapeutic efficacy study protocol and all participants were encouraged to come to the health center if they fill any symptoms in non-scheduled day, therefore, they were not lost.

15. How was treatment on different days describing dosage on each day?

Response: Thank you and corrected accordingly.

16. Were recorded Baseline data on socio-demographic and clinical characteristics?

Response: Thank you and corrected. The baseline data were recorded on the case screening form and a summary is presented in the manuscript.

17. Has quality assurance been performed for microscopy?

Response: Thank you, yes. Experienced microscopists with refresher training participated in the study, the microscopy results were re-read by WHO certified microscopy experts and discrepancies were judged by a third reader.

18. What were the criteria for establishing adverse effects?

Response: Thank you, your comment is well appreciated. Adverse effects are established by WHO and clearly stated in the study protocol, and we defined the adverse events in the revised manuscript. Your comment is well appreciated.

19. How was security assessed? What criteria?

Response: Thank you. If I got you mean by security assessed? The study site was selected as the sentinel site by the selection criteria of therapeutic study sites previously. This is a regular follow-up study for monitoring the frontline drugs in the country.

20. Primary Outcomes and secondary? Define each item of this study.

Response: Thank you, your suggestion is well appreciated, primary outcome can be described in different studies as the cure rate of the drug, and secondary outcomes can be defined as the clearance of fever and parasitemia. But we followed the WHO standard therapeutic study protocol and mentioned the response of the study drug one by one as cure rate, parasite, and fever clearance. 

21. Was it not evaluated clearance and presence of gametocytes?

Response: Thank you, The parasite clearance and presence of gametocytes were evaluated and presented in Figure 3.

Results

22. Make a flowchart describing how the selection was made and the number of participants included.

Response: thank you for your suggestion. We thought the total screened, enrolled, and excluded participants are well described on the flowchart (Figure 1).

23. Were participants who did not take CQ as treatment included?

Response: Thank you. No, we enrolled participants who were diagnosed as P.vivax infected and treated by only CQ for follow-up.

24. This phrase "couldn't complete the 28-day follow-up and were therefore excluded", needs to be included in the flowchart.

Response: The reason to exclude from the study is developing hypersensitivity to the drug. this comment is rephrased and appreciated on the revised manuscript and, on the flow chart stated as “hypersensitive with drug”.

25. In figure 2, insert the correlation confidence interval? p value?

Response: thank you and accepted.

26. What purpose of making correlation?

Response: Thank you for your question. Making correlations to know the relation between the age of participant and parasitemia, and measure the strength of the linear relationship.

27. Why was the dosage of chloroquine not performed for classification of recurrence?

Response: Thank you, dosage of treatment was according to national treatment guidelines. Doing the drug blood level is one of the important things to classify the true treatment failure, but even though only had one treatment failure we didn’t perform the test due to lack of resources and we make it a limitation.

28. Why were these data (K-M analysis) not presented through the survival curve?

Response: Thank you, we have made the survival curve but, the curve do not show considerable information due to only one treatment failure on day 28, as explained in table form.

29. what was the parasitemia of the recurrent case? Put in the figure 3.

Response: Thank you for your suggestion, the parasitemia of the recurrent case is 1760/µ and included in figure 3 as suggestion. 

30. This study cannot speak about resistance to CQ, as resistance was not confirmed in these participants.

Response: Thank you, yes this study reported the resistance monitoring activity results in the study area. 

31. What was done to confirm resistance?

Response: Thank you, we have reported clinical failure resistance was not confirmed due to unable to measure the drug’s blood level stated as a limitation. 

Discussion

32. Relapse, malabsorption of drug, poor drug quality and re-infection could be contributing factors to treatment failure of P. vivax? Insert into the discussion on this aspect.

Response: thank you, we have appreciated your suggestion, and corrected it accordingly.

33. drug quality control was not performed?

Response: Thank you, The study drug was provided by the world health organization (WHO) as quality checked with responsibility and we have not been mandated to do drug quality.

34. Better describe all limitations: sample size, not having performed chloroquine dosage, not performing drug quality control

Response: Thank you. Thank you, comments are accommodated where appropriate.

---

## [Decision Letter · Decision Letter 1]

14 Oct 2022

PONE-D-22-18161R1Therapeutic efficacy of Chloroquine for the treatment of uncomplicated Plasmodium vivax infection in Shewa Robit, Northeast EthiopiaPLOS ONE

Dear Dr Temesgen,

Thank you for submitting your manuscript to PLOS ONE. After careful consideration, we feel that it has merit but does not fully meet PLOS ONE’s publication criteria as it currently stands. Therefore, we invite you to submit a revised version of the manuscript that addresses the few points raised during the review process. Some few points of the reviewer 2 are necessary, including the exclusion and inclusion criteria; the method used to calculate the previous recurrence, the dose administered to patients was the same or was adjusted by body weight? 

We look forward to receiving your revised manuscript.

Kind regards,

José Luiz Fernandes Vieira

Academic Editor

PLOS ONE

Journal Requirements:

Reviewers' comments:

Reviewer's Responses to Questions

**Comments to the Author**

1. If the authors have adequately addressed your comments raised in a previous round of review and you feel that this manuscript is now acceptable for publication, you may indicate that here to bypass the “Comments to the Author” section, enter your conflict of interest statement in the “Confidential to Editor” section, and submit your "Accept" recommendation.

Reviewer #1: All comments have been addressed

Reviewer #2: (No Response)

2. Is the manuscript technically sound, and do the data support the conclusions?

Reviewer #1: Yes

Reviewer #2: Partly

3. Has the statistical analysis been performed appropriately and rigorously? 

Reviewer #1: Yes

Reviewer #2: No

4. Have the authors made all data underlying the findings in their manuscript fully available?

Reviewer #1: Yes

Reviewer #2: No

5. Is the manuscript presented in an intelligible fashion and written in standard English?

Reviewer #1: Yes

Reviewer #2: Yes

6. Review Comments to the Author

Reviewer #1: Dear

This study made the appropriate corrections recommended during the correction process. Therefore, I suggest accepting the study.

Reviewer #2: 1. Inclusion criteria not clear: needs to be made clearer, especially like other WHO criteria

2. Exclusion criteria: needs to be made clearer, especially like other WHO criter

3. Make it clear that previous recurrence calculation

4. State that all participants take the same treatment dose

5. How safe was the treatment? What assessment for this security?

6. Primary and secondary definition.

7. Why were these data (K-M analysis) not presented through the survival curve? The survival curve serves exactly to see the recurrence time because in the table it is not clear. Table 3 is not adequate.

7. PLOS authors have the option to publish the peer review history of their article (what does this mean?). If published, this will include your full peer review and any attached files.

Reviewer #1: No

Reviewer #2: No

---

## [Author Response · Author response to Decision Letter 1]

24 Oct 2022

Response to reviewers

Point-by-point response to reviewer #2

Comments to the Author

1. If the authors have adequately addressed your comments raised in a previous round of review and you feel that this manuscript is now acceptable for publication, you may indicate that here to bypass the “Comments to the Author” section, enter your conflict of interest statement in the “Confidential to Editor” section, and submit your "Accept" recommendation.

Reviewer #1: All comments have been addressed

Reviewer #2: (No Response)

2. Is the manuscript technically sound, and do the data support the conclusions?

Reviewer #1: Yes

Reviewer #2: Partly

3. Has the statistical analysis been performed appropriately and rigorously?

Reviewer #1: Yes

Reviewer #2: No

4. Have the authors made all data underlying the findings in their manuscript fully available?

Reviewer #1: Yes

Reviewer #2: No

5. Is the manuscript presented in an intelligible fashion and written in standard English?

Reviewer #1: Yes

Reviewer #2: Yes

6. Review Comments to the Author

Reviewer #1: Dear

This study made the appropriate corrections recommended during the correction process. Therefore, I suggest accepting the study.

Response: Thank you

Reviewer #2: 1. Inclusion criteria not clear: needs to be made clearer, especially like other WHO criteria

Response: thank you, we have corrected it accordingly.

2. Exclusion criteria: needs to be made clearer, especially like other WHO criter

Response: thank you, corrected accordingly.

3. Make it clear that previous recurrence calculation

Response: thank you for your suggestion, we didn’t do a recurrence determination (recrudescence vs reinfection) that we tried to describe as a limitation of the study. 

4. State that all participants take the same treatment dose

Response: thank you, the dose administered to the study patients was adjusted by body weight, which we made clear on the MS. 

5. How safe was the treatment? What assessment for this security?

Response: thank you, CQ is a drug used on the national P.vivax treatment guideline and the drug was provided by WHO for the study, no need of doing additional analysis. On the follow-up visit, we assessed the adverse events by direct questioning and using adverse event recording formats. 

6. Primary and secondary definition.

Response: thank you, primary and secondary treatment outcomes are defined based on WHO treatment outcome definition and we defined the outcomes in the MS on the study endpoint by considering your suggestion. 

7. Why were these data (K-M analysis) not presented through the survival curve? The survival curve serves exactly to see the recurrence time because in the table it is not clear. Table 3 is not adequate.

Response: thank you, corrected accordingly.

---

## [Editor Report · Decision Letter 2]

26 Oct 2022

Therapeutic efficacy of Chloroquine for the treatment of uncomplicated Plasmodium vivax infection in Shewa Robit, Northeast Ethiopia

PONE-D-22-18161R2

Dear Dr. HABTAMU Belay Temesgen,

We’re pleased to inform you that your manuscript has been judged scientifically suitable for publication and will be formally accepted for publication once it meets all outstanding technical requirements.

Kind regards,

José Luiz Fernandes Vieira

Academic Editor

PLOS 

Reviewers' comments:

All the suggestions were done by authors.

---

## [Editor Report · Acceptance letter]

22 Nov 2022

PONE-D-22-18161R2 

Therapeutic efficacy of Chloroquine for the treatment of uncomplicated Plasmodium vivax infection in Shewa Robit, Northeast Ethiopia 

Dear Dr. Temesgen:

I'm pleased to inform you that your manuscript has been deemed suitable for publication in PLOS ONE. Congratulations! Your manuscript is now with our production department. 

Kind regards, 

on behalf of

Dr. José Luiz Fernandes Vieira 

Academic Editor

PLOS ONE